# The Management of Agriculture Plastic Waste in the Framework of Circular Economy. Case of the Almeria Greenhouse (Spain)

**DOI:** 10.3390/ijerph182212042

**Published:** 2021-11-16

**Authors:** Francisco José Castillo-Díaz, Luis Jesús Belmonte-Ureña, Francisco Camacho-Ferre, Julio César Tello-Marquina

**Affiliations:** 1Research Centre CIAIMBITAL, Department of Agronomy, University of Almería, 04120 Almería, Spain; Franjcd95@hotmail.com (F.J.C.-D.); fcamacho@ual.es (F.C.-F.); jtello@ual.es (J.C.T.-M.); 2Research Centre CIAIMBITAL, Department of Economy and Business, University of Almería, 04120 Almería, Spain

**Keywords:** circular economy, waste management, agriculture, plastic waste, sustainable development

## Abstract

In recent decades, ecosystems have suffered diverse environmental impacts caused by anthropogenic activities, including the dumping of plastic waste. This situation has prompted the European Union to introduce a new policy based on the circular economy. In this study, the present state and future perspectives on the generation and treatment of plastic waste in the intensive agriculture of Almeria (Spain) are analyzed. This activity generates 1503.6 kg·ha^−1^·year^−1^, on average, of plastic waste with an approximate treatment cost of 0.25 €/kg. The present study shows that the volume of plastic waste from intensive agriculture in Almeria is constantly increasing (48,948.2 tons in 2020/21) and it is suggested that the current management system does not meet the needs of the sector. Although it presents great opportunities for improvement under the framework of the circular economy. Furthermore, this work reports a direct relationship between the price of the raw materials needed for the production of plastic and the volume of recycled plastics. For this reason, it would be advisable for the administration to consider the implementation of a tax rebate system for the sector and specifically when the petroleum derivatives used to manufacture plastic are less expensive, and the recycling option is not so attractive.

## 1. Introduction

Anthropogenic activities have deteriorated natural ecosystems while compromising their sustainability [1,2]. Among these activities, the production of food and fiber required by humans has led to the transformation of ecosystems into agricultural systems [3], which alters their functionality and organization [4]. These parameters have changed with every agricultural revolution triggering an increase in crop yields [5,6,7,8,9,10]. This has resulted in an ever-increasing consumption of natural resources [11]. The metamorphosis of agricultural systems has led to a shift from traditional subsistence agriculture to commercial agriculture, especially in the most economically developed countries [12]. This dynamic is causing serious environmental, social, and economic imbalances in some cases [2,13,14,15,16]. However, despite the changes that have taken place, agriculture continues to have a family-based structure in most countries and it is a major source of employment as well [17].

Agricultural evolution has led farming systems to demand inputs that are linked to plastic polymers [18,19,20,21,22]. The characteristics of plastic elements make them one of the primary pollutants in ecosystems, mainly due to their tendency to fragment into small particles, such as microplastics and nanoplastics [23,24,25,26,27,28]; and by the emission, due to ultraviolet radiation from the sun, of some chemical substances that are added during the manufacturing process (additives) to improve the characteristics of the plastic (e.g., biophenol) [29,30]. Unfortunately, the dumping of plastic waste is a common practice and marine ecosystems have experienced the highest incidence of dumping [31,32], which affects their flora and fauna [25], and its associated microbiome [33]; as well as their physical environment [34]. Negative externalities have also affected terrestrial ecosystems and agriculture has become a source of pollution, particularly in the case of intensive agriculture due to the massive use of inputs made from plastics [18,22]. Thus, poor management of this material can lead to the accumulation of microplastics in soil profiles [35] or to the internal infestation of some animals intended for human consumption when fed with agricultural biomass [36], can lead to the accumulation of polymers and chemicals in plant and animal tissues that can be transferred to the food chain [30,37,38]. This may pose a risk to the health of living beings, including humans [26,30,39]. For example, some additives added to plastics have been observed to act as endocrine disruptors (e.g., biophenol) [29,30].

In most cases, the lack of sustainability generated by the conventional agricultural production process contravenes the United Nations (UN) principles for “Sustainable or Lasting Development” formulated in 1992 [40]. In September 2015, the UN approved the 2030 Agenda for Sustainable Development addressing the economic, social, and environmental aspects of greatest global interest [41], which includes the sustainability of agricultural systems [42]. The Agenda has guided the development of various strategies and regulatory bases of its members. In addition, the European Union (EU) proposed the European Green Pact as the central axis of this change to implement a long-term production model based on sustainability [43] by replacing its current linear economic model with an alternative model, such as the circular economy (CE). In 2015, the foundations were laid for a European Circular Economy Action Plan to promote the reduction, reuse, and recycling of inputs used in the various production systems to minimize the waste generated. It details seven lines of action to address the problem generated by waste, which includes plastics [44]. This action plan underwent a revision in 2020 and the lines of action proposed in the previous plan have been completed [45]. These new measures aim to reduce the pollution caused by plastics and they include a ban on single-use plastics, which affects the materials used in food marketing [46]. In addition, the EU has recently reformulated its waste management regulations affecting the treatment of agricultural plastics (hazardous and non-hazardous) [47,48,49,50]. These regulations make special mention of the reuse of waste generated in production processes by transforming it into by-products.

Intensive agriculture under plastic, or in greenhouses, is very important on a European scale. More than 43% of the greenhouse area in the world is located in the EU, i.e., about 175,000 ha of the 405,000 ha identified worldwide [51]. Spain, France, Greece, Italy, and the Netherlands are the areas of greatest significance and Spain has the largest area of protected cultivation with 71,783 ha. Protected agriculture is not homogeneously distributed in the country but rather concentrated in agricultural areas [52]. Thus, the province of Almeria is the largest agricultural region of protected crops in Spain reaching 32,554 ha of greenhouses in 2020. These are mostly concentrated into two agricultural regions (Campo de Dalías and Bajo Andarax—Campo de Níjar) [53] that are socioeconomic pillars in the territory [54,55]. Despite being systems that typically consume large amounts of resources [51], they are known for their efficient use of these resources thanks to the technology involved in different production processes [56]. However, they also generate high amounts of different kinds of waste [18,22,57], plastics being one of them [21,22]. For some years now, the Food and Agriculture Organization of the United Nations (FAO) has recommended a series of practices in protected agriculture with the aim of mitigating climate change and promoting the sustainability of the systems [56]. This has been coupled with an effort made by the EU to promote the establishment of “circular horticulture” in agriculture under plastic, which has expanded significantly in its territory [51]. Indeed, the intensive greenhouse horticultural sector has been described as a suitable place for its implementation [57,58,59,60,61]. However, due to the recent introduction of various strategies and action plans, the establishment of circular horticulture is considered one of the biggest challenges of the agricultural systems, including those located in the province of Almeria [62].

Due to the socio-economic importance of greenhouse agriculture in Almeria and its huge need for plastics, it is essential to identify the opportunities of the agricultural system to obtain its production under the principles of CE. Thus, the objectives of this work have been the following:To calculate the production of agricultural plastic waste or by-products in the greenhouse fruit and vegetable production system in Almeria along with the current management system.Identify and qualitatively assess the various existing alternatives for the management of agricultural plastic waste along with the opportunities supported by the current regulatory framework that may exist for farmers and managers.To evaluate the relationship between the price of oil and the amount of recycled plastic in the member States of the EU.

## 2. Materials and Methods

### 2.1. General and Specific Stages of the Research Process

For this research, a compilation, classification, verification, and detailed analysis of the information obtained from various studies, technical reports, regulations, and statistics from various public and private organizations was carried out. These organizations had authority and/or relationships with the subject matter addressed and these ranged from local to international. In addition, telephone consultations with numerous departments of regional and local public institutions in charge of managing the data of interest were made to validate the information gathered from their official web pages. Telephone or e-mail interviews were conducted with different agents involved in the production, marketing, or management of agricultural plastics with the aim of expanding and verifying the information found in the bibliography.

The analysis of the literature on plastic waste/by-products was carried out using the Scopus and Web of Science (WoS) databases. This procedure has already been used in other studies related to waste management in agriculture [58] resorting in some cases to the use of specialized software for information processing (Excel Microsoft 365, SigmaPlot 14.0 and SPSS Statistics v.26). Table 1 shows the specific stages of the study.

### 2.2. Details of the Statistical Treatment Performed for Objectives 1 and 2

After obtaining the data and in view of the variety of publications found on the subject analyzed, the information was processed by synthesizing it into homogeneous groups (i.e., element) as shown by other authors [18,22] and the technical documents of the Reinwaste project [63,64,65,66,67] to ease the representation of the production and management of waste/by-products with the aim of obtaining a single value.

Estimates for Objective 1 were made for commercial “scratch and shake”, flat vine and multi-tunnel greenhouses where horticultural crops were grown under the common practices of the area (conventional and organic crops grown on natural soil), from the literature identified in different databases [18,22,63,64,65,66,67]. The data contained in the literature consulted were obtained from an average of plastic waste generated in greenhouses with a surface area of 0.15–1 ha. In these documents, the total production of plastic waste is presented, without indicating the standard deviation. The data are presented in kg-ha^−1^. Finally, the total production of plastic waste per unit area was also calculated and the total production for the total greenhouse area in 2020 was estimated, according to data provided by the regional government [53].

In order to describe the current plastic waste management system in the province of Almeria, direct consultation with the key agents involved in the process (companies) was used. After analyzing the information collected, the plastic waste managers were classified according to the services they offered. Finally, the mean value and standard deviation of the incentives offered to farmers were calculated, as well as the total cost of plastic waste treatment (€/kg).

The estimate of the average price of plastic waste management was found through the average cost for service, including transportation (€/kg). Finally, the estimate made in Objective 2 for the cost of alternatives was made through information available in the literature (cost + dose) [68,69], the technical documents of the Reinwaste project [63,66,67] and consultation in specialized supply centers to reach the cost increase in €/ha.

### 2.3. Additional Details of the Methodology for Objective 3

After obtaining the values of the Brent crude oil price per barrel (C), the average annual value was calculated. This value reflects a change in the variable from the time of origin (t) to a monthly lag of twelve months before and after this value (t − 12 to t + 12) to observe the influence of the variable on the annual proportion of recycled plastic (%). The results were expressed in the unit of origin ($/barrel). For this purpose, the following expressions were used:(1)$$\bar{C_{t-n}}=\frac{\sum_{t}^{n}C}{12}$$
(2)$$\bar{C_{t+n}}=\frac{\sum_{t}^{n}C}{12}$$



C¯:average price of a barrel of crude oil ($barrel)





t:time (months)



n: number of months of lag time (−12 ≤ t ≤ 12)

The correlation between the values was obtained using Pearson’s coefficient (r) with the F test to check the assumptions necessary for its application. The dependent variable was the amount of ‘packing’ plastic recycled (%) and the amount of ‘light packaging’ recycled in Almería (plastic, briks and cans; %) and the independent variables were the different annual prices of a barrel of crude oil (t − 12–t + 12; $/barrel). The 27 Member States were then subdivided into two groups (Table 2). The first group included the countries that showed an average Pearson coefficient (t − 12–t + 12) higher than the EU average (G1). The second group included the territories whose average Pearson coefficient (t − 12–t + 12) was lower than this average (G2).

## 3. Results and Discussion

### 3.1. Characterization and Estimation of the Production of Agricultural Plastic Waste in the Productive Sector of Greenhouse Agriculture in Almeria

Plastic is linked to most of the components and techniques used in the production process under protection in Almeria (Table 3). It is present in the covering and natural ventilation systems of greenhouses, irrigation equipment and active climate control, agricultural inputs and disinfection processes. It is also used in auxiliary structures such as irrigation ponds and the marketing of the goods [18,22,59,60,61,63,64,65,66,67,70,71,72]. The determination of the generation period of this tyspe of waste is established by the crop cycles with a productive seasonality due to their intrinsic characteristics [73,74] Specifically, it was estimated that around 90% of agricultural plastic waste is generated in the province of Almeria between August and September [18].

In detail, it was estimated that agriculture under plastic in the province of Almeria generates an amount of plastic of 1503.6 kg·ha^−1^·year^−1^ in the production of fruits and vegetables (an annual amount in the system of 48,948.2 t) (Table 3). The amount of waste from covering elements, soil disinfection, and double roofs of greenhouses can account for up to three-quarters of this amount, the rest being made up of the different inputs used in the agricultural system. The packaging of phytosanitary products was classified as hazardous waste while other waste was not classified as such [47,48,75,76]. The increase in the surface area registered in recent years [53,58] would cause an increase in plastic waste or by-products generated in the production process, which is concentrated in a few agricultural districts due to the distribution of the greenhouses. Ninety-one percent of the greenhouse surface area is located in the Campo de Dalías and Bajo Andarax—Campo de Níjar [53].

However, as in other models of greenhouse agriculture, these elements were formed from seven groups of polymers, such as low (LDPE and LLDPE) and high density polyethylene (HDPE), polypropylene, ethylene-vinyl acetate (EVA), polystyrene and polyvinyl chloride (PVC) [20,21,22,77,78,79]. The different types of polyethylene (LDPE, LLDPE and HDPE) and polypropylene made up the largest tonnage of production (96.9%) (Figure 1). The microplastics obtained from the fragmentation of these polymers cause the greatest impact on the *Posidonia oceanica* meadows found on the Almeria coast. Its ecosystem shows a significant increase in the deposition of microplastics from greenhouse agriculture. The damage caused is not only due to the fragmentation of the material, but also to the presence of various additives in plastic materials, such as benzyl phthalate [80], which can cause harmful effects on the health of organisms [81].

However, due to the density of the compounds, the volume occupied by polystyrene increased significantly from 0.4% of the mass to 25.6% of the volume of plastics generated. This material is mainly generated in the raising of seedlings used in agriculture since it is the main component of the trays used in seedbeds. Thus, the Almeria production system generates between 10.3 and 15.8 g of plastic per kilogram of product (not including the containers used in marketing), which is less than the figure of 20 g observed in other sheltered systems [21].

### 3.2. Management Strategies Used for the Treatment of Plastic Waste from Greenhouses: The Case of the Province of Almeria

The processing of the plastic waste generated in greenhouses in the province of Almeria has raised significant problems throughout the life of the greenhouse model mainly due to the seasonality of waste production, an increase in the greenhouse surface area, the difficulty to manage some plastic waste or by-products, and the bad disposal practices of some farmers [18,63]. In 2001, the dumping of agricultural waste in natural areas of Andalusia resulted in a health crisis that required a cleanup of the countryside in the region of Almeria. This crisis revealed the need to implement an adequate agricultural waste management and treatment system in the province of Almeria in light of the expansion of its intensive agriculture. Since then, competent public institutions have developed different financing lines and agreements for initiatives aimed at the massive collection of agricultural waste. They have also developed urban and rural cleanup plans, along with a general program to improve the integrated management of agricultural waste [18,82]. Despite all efforts, the poor management of agricultural plastic waste or by-products from protected agriculture in the provinces of Almeria, Granada, and Huelva, and the indiscriminate abandonment of these leftovers in natural spaces forced the Government of Andalusia to carry out a campaign to collect greenhouse plastic waste [83,84,85] in 2018.

Currently, the province of Almeria has more than 25 treatment and recovery centers for hazardous and non-hazardous agricultural plastic waste distributed throughout its territory, most of them located in the western part of the province. In each of these, the storage, waste recovery, or recycling procedures authorized by competent authorities are applied [86]. The necessary decontamination processes are also used for hazardous by-products [76].

Of the farmers who grow crops under shelter in these centers, 96.2% manage plastic waste and 94% are involved in hazardous product packaging [87]. Only 10% of the management centers handle most of the plastic waste generated on farms and they offer a specialized management service. The agricultural plastic waste most readily accepted by the management plants were polyethylene boxes and containers (80%), and ventilation nets (70%), followed by pipes and thermal blankets (60%) (Table 4). In contrast, none of the management centers dealt with gloves, plastic hives, personal protection suits, or chromotropic traps, perhaps as a result of their specialization. Unfortunately, only 30% of the waste management companies accepted the structure, solarization, and double roof plastics, despite the fact that these are the largest fraction in the amount of waste generated in the intensive system. There is a small number of waste management companies handling containers of phytosanitary products and fertilizers used in agriculture. Although management has improved significantly with the implementation of the container take-back system (SIGFITO), to which most manufacturers adhere to, there is still a significant amount of containers that are not delivered to a specific disposal or treatment point [88]. Since 2013, some fertilizer producers have voluntarily joined this integrated treatment system [76], despite not having the regulatory obligation to implement a collective management model or an individual system until 31 December 2024 [48,76].

The farmer pays the management and transport costs of plastic waste. The maximum cost of these services has been estimated at approximately 0.25 € for each kg of plastic waste produced as shown in Table 4. Regarding the integrated packaging management system, the cost is incorporated into the sale price of the input [88], while for other types of residue payment is made when the waste is delivered to the authorized disposal facility. Some operators have established economic incentives to motivate the delivery of waste (0.03–0.15 €/kg), although the measure is selective in nature and applies only to some types of waste (mainly from greenhouse coverings, irrigation systems, boxes, and containers).

Subsequently, waste managers commercialize the plastic pellets resulting from the physical recycling processes (extrusion) of these materials [89,90]. However, not all of the by-product obtained meets the quality criteria required by this process, so the remaining fraction can be used for energy recovery due to its high calorific value [64,91,92,93,94]. This last method is not considered a recycling protocol [47,50,95]. Some containers of phytosanitary products and fertilizers, generally those with a capacity greater than 200 L, are reused by the companies of origin [96]. It should be noted that some authors recommend that urban waste treatment services manage gloves and chromotropic traps [63,65,66,67], although this measure leads to a regulatory problem because these residues are defined as non-municipal agricultural waste, so this service falls outside of their range of responsibility [97]. In 2018, in the entire Region of Andalusia, only 40.8% of agricultural plastics were recycled. A total of 43.6% was recovered for further treatment and 15.6% was stored for further recovery [98].

Despite the efforts made by the Andalusian Government, the current plastic waste management model does not offer a comprehensive solution for the sector. The selective nature of the management services available forces farmers to resort to multiple entities, which can confuse the producer and also increase transportation costs. It should be noted that the management centers do not make their rates public, thus increasing the number of preliminary steps to be taken by the farmer. For this reason, some associate or commercial entities facilitate this action with a list containing the most relevant information for the management of agricultural plastic waste, including cost, although this is not a widespread practice. On the other hand, some management centers place obstacles in the treatment of certain types of plastic waste (mainly solarization or mulching plastics) due to the dirt adhered to the plastic, an advanced state of degradation, or the mixture with other types of plastic waste. Therefore, it is recommended that the delivery conditions be agreed upon beforehand between the parties involved [20].

Further, the attitude of some farmers can negatively influence the pre-delivery process. It has been reported that a minority of farmers, often older ones, do not perceive plastic management as critically important for the environment and society [21]. This could lead to the relaxation of sorting and cleaning operations for plastic crop residues, and thus make their treatment more difficult [20].

### 3.3. Opportunities for Waste Management and Use of Plastic by-Products by the Different Agents Involved: Inclusion in the CE

In recent years, a strategic framework has been developed and accelerated to promote, prioritize, and favor the recovery of plastic by-products generated in agricultural systems. The action plans have been included under the circular economy strategy [44,45,99], so it is required to establish a circular system that performs an adequate management of plastic waste. Therefore, it is also necessary to favor measures that increase the generation of by-products, the introduction of biodegradable plastic, the reduction of plastic needs, or the adequate management of the waste generated.

Different compostable and biodegradable plastic polymers have been recently developed for agricultural use. Unfortunately, their technological limitation only allows their use in a few productive activities, mainly to replace traditional composts in mulching and mulching raffias. Its characteristics enable it to withstand production cycles of up to nine months, which makes it an ideal replacement for mulching used throughout the crop cycle [59,65,66,69]. Other types of biodegradable mulching, such as straw or rice husks, could also be used [68,69]. Biodegradable plastic mulches have shown variable effectiveness compared to conventional polyethylene material when used in soil disinfection by the solarization technique [100,101]. This could be due to the higher permeability of the compound due to its inability to retain heat and moisture [100]. Repeated monoculture can cause reduced plant growth due to soil fatigue [102,103], which increases the need for plastic and disinfection of the agricultural system to avoid significant decreases in production due to the widespread use of solarization in greenhouse agriculture in Almeria [104]. The addition of organic matter [102] or crop diversification [103] can prevent this phenomenon, thus reducing the need for plastic. Some authors have advocated for agricultural biomass as an alternative for organic amendments [57,60,102] or the cultivation of papaya (*Carica papaya*) as an option to decimate the low pluralization of plant species used in the Almeria model [105]. Likewise, the use of alternative plastics, biodegradable polymers in items such as chromotropic traps, trellising rings [22] or flower pots [106,107], or broader reuse in other agricultural inputs sucha s the packaging of phytosanitary products and fertilizers [18] could also be considered. The implementation of this final option would largely depend on the resistance of the material due to the toxicity of the formulations that must be contained. The use of ultrasound emitters instead of adhesive traps [22] or a ring-free trellising methodology, where the plants are only guided through the trellising raffia [60], has also been considered.

The sale price of compostable and biodegradable materials represents a considerable increase with respect to some traditional options (Table 5), but these materials facilitate the management of plastic by-products by the farmer or the management plant. On the other hand, rice husk mulching, incorporated through mechanical banding, and biodegradable raffia have been the alternatives with the lowest price increase. There has been a rise in the number of subsidies to encourage the use of biodegradable and compostable plastic, mainly in mulching and trellising elements. In 2021, farmers who were not affiliated with a Fruit and Vegetable Producer Organization (FVPO) had access to a subsidy for the replacement of mulching raffia of 419.29 €/ha [76,108]. Farmers who were affiliated with an FVPO obtained 50% off their invoice in replacing plastic mulch and 66% off the replacement of trellising elements [109,110]. This has solved a large portion of the cost overrun [111].

The use of biodegradable plastic, which decomposes naturally [109], or other organic materials (straw, rice husks, etc.) enables farms to self-manage by incorporating them directly into the soil as raw materials for vermicompost or other types of organic fertilizers. These can act as a part of the nutrient source necessary for the crop [107,112,113] as one of the organic components used in soil biodisinfection processes (i.e., biofumigation or biosolarization) or other agricultural by-products, such as biomass [60]. In fact, the use of alternative materials would facilitate the external management of the biomass or the self-management of the material by the producer by separating any type of plastic from the plant remains [66]. This dynamic would reduce labor by not having to separate the trellising elements from the plant remains. In addition, the inoculation of the soil with earthworms has been recommended to facilitate the proliferation of the microbial flora of the soil by facilitating the biodegradation of the plastic [112]. This practice could provide other benefits as several studies have shown that earthworms improve the physicochemical properties of the soil (improvement of infiltration, porosity, structure or nitrogen mineralization, among others), which allows better crop development [114,115]. On the other hand, organic mulches, such as those formed by straw, offer partially analogous benefits. Their use also leads to an increased organic carbon content [68] and the richness of the fungal community in soil [113], although they should be combined with other amendments or nitrogen compounds to avoid an excessive increase in the carbon/nitrogen ratio.

Unfortunately, no type of compostable or biodegradable plastic whose characteristics meet the needs of resistance and transmissibility demanded by the sector has been commercially developed for the structure of greenhouses, although there are public initiatives to promote its proliferation [116]. Nonetheless, the extension of the useful life of roofing plastics, which is already an ongoing study, could be considered to minimize the generation of waste [117].

Despite the available alternatives, the majority of plastic waste or by-products generated in the greenhouse production system in the province of Almeria must be managed through authorized treatment plants. For this reason, measures such as the establishment of a homogeneous system of incentives to encourage the delivery of waste or a traceability system that allows the identification of those producers who carry out plastic spills should be implemented. A control system has been analyzed by public institutions that is based on physical documentation through a waste management contract and an identification document, or through a documentary record using the compulsory operating logbook in which the information on waste management is entered. This system would facilitate the identification of producers who dump waste [67] and should therefore be introduced as a mandatory measure.

After processing the plastic by-product, the waste management companies obtain a material (plastic pellets) that does not meet the quality criteria demanded by the industry in charge of manufacturing plastic greenhouse coverings. The properties of the recycled raw material are far from those of the basic virgin material [118] and its use would considerably increase production costs while requiring a greater amount of additives and slowing down production process [119]. In some cases, it would also reduce the transmissibility of the cover due to the use of plastic pellets obtained from dyed by-products (e.g., black insect-proof netting). Recycled plastic pellets are marketed at a lower price than virgin material and are destined for a market whose demand is expanding due to current European social and political dynamics [46,47,48,120] that represent an advantageous opportunity for waste managers [89,120]. The main destination of these is usually the production of some agricultural elements (e.g., planters) or urban furniture (e.g., outdoor furniture, garbage containers) [118,121], although they could also be used for industrial purposes, drums or household items, among others [90,122]. Due to the requirements of physical recycling, a portion of the plastic by-products could be used to obtain energy through energy recovery [92], but unfortunately this protocol causes the emission of large quantities of greenhouse gases, which leads to questions about its sustainability. Current lines of research have investigated the potential of some larvae of tenebrionids (*Tenebrio molitor*) or lepidoptera (*Galleria mellonella*) to decompose some plastic polymers, such as polyethylene [123,124,125], but this technology is still under development.

Finally, the parties involved (administration, managers, and producers) should promote and implement the necessary actions to help improve the current system of managing agricultural plastics. These actions should include encouraging the introduction of alternative materials, promoting environmental awareness campaigns to inform producers about the pre-treatment of plastic by-products (management plants, type of service performed, and cost of the service), and a traceability system that allows the identification of producers who discharge waste. For the latter two measures, information technology can be used through a mobile app as well as expanding the incentive system to all plastic waste and by-products generated to offer an amount close to 0.25 €/kg. The last measure is recommended by the EU to encourage the recycling of agricultural plastic waste [99].

### 3.4. External Influence of the Cost of Oil on the Management of Plastic by-Products: Relationship between the Cost of a Barrel of Brent Oil and the Percentage of “Packing” Plastic

A direct correlation (*p* ≤ 0.05; *p* ≤ 0.01; *p* ≤ 0.001) was observed between the price of a barrel of oil and the proportion of recycled plastic (Figure 2) from 1997–2018, although the relationship of the variables depended on each particular European territory.

Thus, the countries belonging to group G1 achieved the highest ratio coefficients with average values ranging from 54.5% to 67.7% (Figure 2). These values correspond to the ratio at t + 12 and t − 11, respectively. Ireland was the member that showed the highest ratio within this group of territories at 87.9% (in t + 4). Spain reached a ratio that ranged from 47.9–64.1%, while the province of Almeria has a ratio of 36.7–1.0%.

Member states included in G2 obtained a lower ratio than the European average. As in the case of G1, the highest Pearson’s coefficient tended to occur in the range between t − 12 and t. However, from t − 4 and beyond, the relationship acquired a negative sign (Figure 2). Thus, countries such as Poland, Germany, and Latvia showed an inverse correlation throughout the range of study. Latvia reported the lowest relationship at t + 12 with −51.3%. The difference shown between the different EU Member States in the ratio could be a result of a disparate application of their plastic waste management and treatment policies [126] where some of the territories would have decoupled their recycling rate from the cost of the raw material.

The results suggest that the Brent price of oil could be an influential variable in the annual planning of European plastic waste management despite targets set by the EU for plastic waste management, which requires its member states to increase the treatment of plastic waste [47,75]. Thus, recycled plastic raw material is offered to the market at a lower cost than that obtained from virgin products [127]. The demand for crude oil from the plastic manufacturing sector [128] could be reduced during periods when the cost per barrel of oil remains high by substituting it with recycled plastic material [129]. However, the relationship observed in Figure 2 also poses difficulties in the face of the volatility shown by the price of crude oil. The increased need for additives in the manufacture of plastic and the slowing down of the production process due to the use of recycled plastics [118] may lead to a sharp drop in demand for the by-product in periods when the price of oil remains low, thus limiting the interest of processing plants to apply a recycling treatment to the polymers delivered. The material could just be sorted, to increase the stock and wait for an increase in the demand for by-products; be used for other purposes, such as energy recovery (which is not legally considered a recycling protocol); or be accumulated in landfills in those European countries where it is allowed [47,48,130,131,132]. In this case, the Administration, within the framework of the CE, must articulate some compensation system that makes the demand for plastic by-products attractive and maintains the demand of manufacturers for the production of new polymers based on recycled material.

## 4. Conclusions

Greenhouse fruit and vegetable agriculture in the province of Almeria uses large amounts of plastic in its productive sector. The annual estimate of this fraction was 1503.6 kg·ha^−1^·year^−1^, which consists of waste and by-products of materials of different origins, but also where the plastics of structure, strips, solarization, and double roof accounted for almost three-quarters of the total. The upward trend in the greenhouse area prompts us to reflect on an increase in the production of waste and by-products in the Almeria model.

Unfortunately, the results of this research suggest that the current system of external management of plastic waste and by-products does not offer a complete solution to the needs of the sector. Despite the significant progress made, plastic dumping still occurs in the natural areas of the province where a negative influence can be seen from all the different parties involved. Nevertheless, the Almeria model would exhibit high potential for improvement since its characteristics seem to facilitate its inclusion within the principles of CE production and thus increase its environmental sustainability. Therefore, it is necessary to facilitate the inclusion of alternative materials and implement the available measures by the Administration to correct the situation, such as establishing a mandatory traceability system in the sector or establishing a system of incentives and compensation to make attractive the delivery of plastic by-products to the manager by the farmer and the demand for recycled plastic chippings by farmers for the production of new materials, respectively. All this under the framework of the EC.

## Figures and Tables

**Figure 1 ijerph-18-12042-f001:**
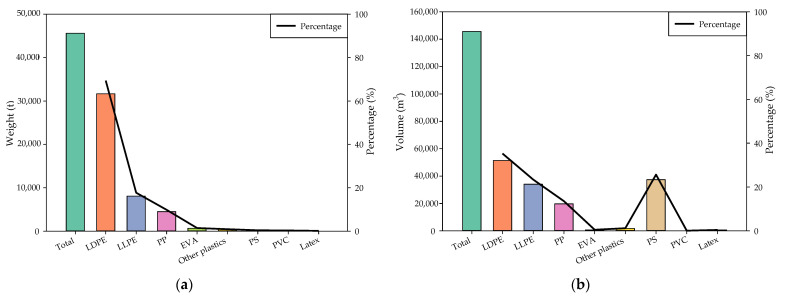
Generation of plastic waste/by-products in the Almeria model subdivided by type of polymer: (**a**) mass estimation; (**b**) volume estimation. LDPE: low density polyethylene; LLDPE: linear low density polyethylene; EVA: ethylene-vinyl acetate; PVC: polyvinyl chloride; HDPE: high density polyethylene; PP: polypropylene; PS: polystyrene. Source: own elaboration based on data provided by other authors [22].

**Figure 2 ijerph-18-12042-f002:**
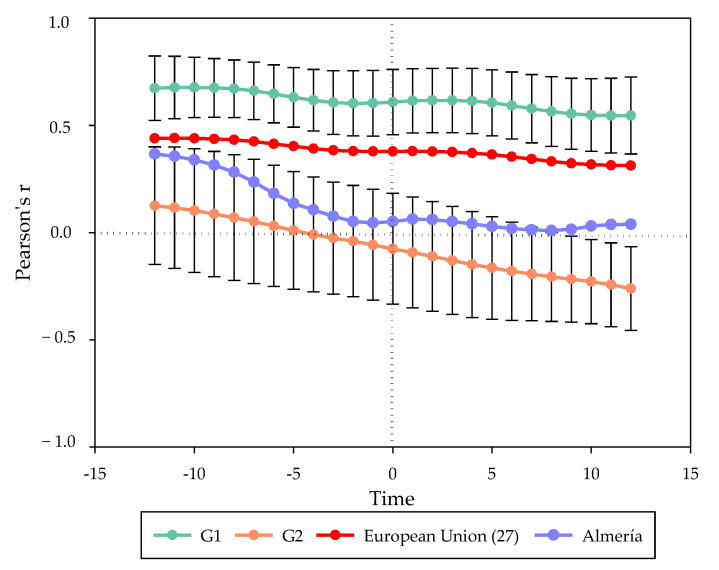
Evolution of Pearson’s coefficient in the “packing” plastic ratio versus the monthly cost of a barrel of oil with a monthly lag from t − 12 to t + 12. Source: own elaboration based on data obtained from Eurostat (consulted on 17 May 2021), EIA (consulted on 17 May 2021) and CAGPDR (consulted on 2 November 2021).

**Table 1 ijerph-18-12042-t001:** Specific methodology and sources of information consulted.

Objetives	Procedure	Source of Information
1. To estimate the production of agricultural plastic waste or by-products in the greenhouse fruit and vegetable production system in the province of Almeria, together with the current management system.	Review and analysis of technical studies and scientific research.	Official websites of public agencies: CAGPDR, MAPA and ERDF.
	Contrast of information found by means of the agents qualified to do so.	Websites of public and private research centers: FC and CIAIMBITAL.
	Processing, analysis, and graphic representation of the results (Excel Microsoft 365 and SigmaPlot 14.0).	Scientific literature obtained from search engines such as Scopus and Web of Science (WoS).
2. Identify and qualitatively assess the various existing alternatives for the management of agricultural plastic waste, together with the opportunities that may exist for farmers and managers, which are supported by the current regulatory framework.	Identification of key agents (public and private organizations).	Information obtained from interviews with CA, IEPA, and ES.
	Telephone or email interview with key agents.	Official websites of public and private organizations: CAGPDR, BOE, EUR-lex, MAPA, and SIGFITO.
	Review and analysis of technical studies and scientific research. Processing, analysis, and graphical representation of the results (Excel Microsoft 365 and SigmaPlot 14.0).	Websites of public and private research centers: IFAFA, CIAIMBITAL, ERDF, and FC.
	Review and analysis of regulatory bases and strategies at local, regional, national, and international levels.	Scientific literature obtained from search engines such as Scopus and Web of Science (WoS).
3. To evaluate the relationship between the price of oil and the amount of recycled plastic in the EU Member States.	Review and analysis of official statistics on crude oil barrel price (series 1987–2020), percentage of “packing” recycled plastic (series 1997–2018) (Article 6.1 Directive 94/62/EC) (consulted on 17 May 2021) and percentage of ‘light packaging’ recycled in Almería (consulted on 2 November 2021)	Crude oil barrel price: EIA.
		Percentage of “packing” recycled plastic: Eurostat.Percentage of ‘light packaging’ recycled in Almería: CAGPDR.
	Statistical processing, analysis, and graphical representation of results (Excel Microsoft 365 and SigmaPlot 14.0 and SPSS Statistics v.26).	
		Scientific literature obtained from search engines such as Scopus and Web of Science (WoS).

CAGPDR: Consejería de Agricultura, Ganadería, Pesca y Desarrollo Rural; MAPA: Ministerio de Agricultura, Ganadería y Pesca; FC: Fundación Cajamar; ERDF: European Regional Development Fund, REINWASTE project; FC: Fundación Cajamar; CIAIMBITAL: Centro de Investigación en Agrosistemas Intensivos Mediterráneos y Biotecnología Agroalimentaria de la Universidad de Almería; C. A: Cooperativas Agroalimentarias; IEPA: Engineers in charge of the production process of plastics producing companies; ES: Supply companies; IFAPA: Instituto de Investigación y Formación Agraria y Pesquera; EIA: Energy Information Administration of the United States; Eurostat: European Statistical Office.

**Table 2 ijerph-18-12042-t002:** Subdivision made in the graphical representation of Pearson’s coefficient.

Group	Countries		
G1	Czech Republic	Denmark	Luxembourg
	Spain	Estonia	Malta
	The Netherlands	Ireland	Austria
	Slovenia	Greece	Portugal
	Sweden	France	Finland
	Slovakia	Croatia	
	Belgium	Italy	
G2	Bulgaria	Lithuania	Hungary
	Germany	Romania	Poland
	Cyprus	Latvia	

Source: own elaboration.

**Table 3 ijerph-18-12042-t003:** Estimation of the production and proportion of agricultural plastic waste/by-products by element and total in the greenhouse production system in the province of Almeria. Values (mean ± standard deviation).

Element	Polymer	Production (kg·ha^−1^·year^−1^) (n = 5)	Proportion (%)
Structure plastic	LDPE, LLDPE, EVA	707.8 ± 83.3	47.6 ± 2.9
Solarization plastic	LDPE	224.1 ± 29.0	15.4 ± 3.3
Double roof film	EVA	135.2 ± 19.0	8.5 ± 1.4
Irrigation system pipes	HDPE, PVC	113.6 ± 15.1	7.6 ± 0.7
Geotextile netting	PP	64.4 ^1^	3.9 ^1^
Trellising clips	LDPE	57.9 ± 24.6	4.1 ± 2.6
Trellising raffia	PP	50.6 ± 31.9	3.2 ± 1.9
Chromotropic traps	LDPE	30.4 ± 13.2	2.0 ± 0.8
Ventilation netting	HDPE	23.9 ± 2.9	1.6 ± 0.1
Thermal blankets	LDPE, LLDPE, EVA	15.3 ± 6.0	1.0 ± 0.3
Fertilizers (bags + containers)	LDPE	14.7 ± 5.7	0.9 ± 0.3
Returnable plastic containers	HDPE, PS	14.2 ± 1.0	1.0 ± 0.1
Field boxes	HDPE	13.6 ± 11.0	0.7 ± 0.7
Non-returnable plastic containers	HDPE	9.9 ± 9.2	0.6 ± 0.6
Plastic containers for phytosanitary products	HDPE	9.6 ± 6.7	0.6 ± 0.4
Plastic hives	LDPE	9.2 ± 0.3	0.6 ± 0.1
Gloves	Latex	4.8 ± 4.0	0.3 ± 0.4
Personal protection suit	HDPE	4.0 ± 4.2 ^2^	0.3 ± 0.3 ^2^
Packaging of biological control products	HDPE	0.4 ± 0.3	0.1 ± 0.0
Total	-	1503.6	-

LDPE: low density polyethylene; LLDPE: linear low density polyethylene; EVA: ethylene-vinyl acetate; PVC: polyvinyl chloride; HDPE: high-density polyethylene; PP: polypropylene; PS: polystyrene. N: number of values. ^1^: n = 1; ^2^: n = 2; source: own elaboration based on data provided by other authors [18] and technical documents of the Reinwaste project [63,64,65,66,67].

**Table 4 ijerph-18-12042-t004:** Management offer, incentive system, and possible treatments of waste/by-products in the Almeria model (n = 10).

Element	Management (%) ^1^	Incentive ^1^		No Incentive ^1^			Charge Transportation Costs ^1^	Destination ^2^
		Proportion (%)	Remuneration (€/kg)	No Charge	Charge			
				Proportion (%)	Proportion (%)	Cost (€/kg)		
Structure plastic	30.0	66.7	0.03 ± 0.01	0.0	33.3	0.10	33.3 ^3^	A, B
Solarization plastic	30.0	0.0	-	0.0	100.0	0.14 ± 0.05	0.0	A, B
Double roof film	30.0	0.0	-	0.0	100.0	0.14 ± 0.05	0.0	A, B
Irrigation system pipes	60.0	16.7	0.03	66.7	16.6	0.10	0.0	A
Geotextile netting	50.0	0.0	-	0.0	100.0	0.12 ± 0.01	0.0	A, B
Trellising clips	20.0	0.0	-	0.0	100.0	0.15 ± 0.07	0.0	A, B
Trellising raffia	20.0	0.0	-	0.0	100.0	0.15 ± 0.07	0.0	A, B
Chromotropic traps	0.0 ^6^	-	-	-	-	-	-	-
Ventilation netting	70.0	0.0	-	0.0	100.0	0.13 ± 0.03	0.0	A, B
Thermal blankets	60.0	0.0	-	0.0	100.0	0.11 ± 0.01	0.0	A, B
Fertilizer bags	20.0	0.0	-	50.0	50.0	0.15 ± 0.07	0.0	A, B
Fertilizes containers	10.0	0.0	-	0.0	100.0	0.39 ^4^	-	A, B, C ^5^, D
Plastic containers ^7^	80.0	87.5	0.15 ± 0.03	0.0	12.5	0.10	0.0	A, B
Containers for phytosanitary products	10.0	0.0	-	0.0	100.0	0.39 ^4^	0.0	A, B, C ^5^, D
Plastic hives	0.0 ^6^	-	-	-	-	-	-	A, B
Gloves	0.0 ^6^	-	-	-	-	-	-	-
Personal protection suit	0.0 ^6^	-	-	-	-	-	-	-
Packaging of biological control products	10.0	0.0	-	0.0	100.0		0.0	A
Average management cost	-	-	-	-	-	0.23 ± 0.02	-	-

Source: ^1^: own elaboration; ^2^: own elaboration based on data provided by other authors [18] and technical documents of the Reinwaste project [63,64,65,66,67]; ^3^: only if the amount is higher than 3 t; ^4^: price included in the cost of the phytosanitary or fertilizer; ^5^: only from containers of products that have agreements with SIGFITO; ^6^: the management facilities consulted do not accept this waste. ^7^: included are returnable plastic containers, non-returnable plastic containers, field boxes and flower pots. A: recycled; B: energy recovery; C: deposit in the SIGFITO container; D: reuse. Transport costs: 100–150 €/service.

**Table 5 ijerph-18-12042-t005:** Economic evaluation for the implementation of available alternatives to plastic for the farmer in the agricultural input market.

Input	Alternative	Cost of the Material (€/ha)	Management Cost (€/ha)		Cost Overrun (€/ha)	Subsidy	Cost Overrun after Subsidy
Padding	Conventional	849.6	70.0		-	-	-
	Compostable	2016.0	19.7 ^1^		1116.1	50.0% of the bill	108.1
	Biodegradable	4152.0	19.7 ^1^		3252.1		1176.1
	Straw	1933.3	E	1320.0	2333.7	-	2333.7
			D	810.0	1823.7	-	1823.7
			EM	370.0	1383.7	-	1383.7
	Rice husk	1575.0	E	1320.0	1975.4	-	1975.4
			D	810.0	1465.4	-	1465.4
			EM	370.0	1025.4	-	1025.4
Trellising raffia	Conventional	108.9	13.6		-	-	-
	Compostable	622.2	4.5 ^1^		504.2	419.2 €/ha o 66.0% of the bill	85.1–93.6
	Biodegradable	559.2	4.5 ^1^		441.2		22.1–72.1
Trellising clips	Conventional	130.5	10.2		-	-	-
	Compostable	777.9	6.7		643.9	50.0% of the bill	255.0
	Biodegradable	659.2	6.7		525.2		196.5
Traceability system	Conventional	-	-		-	-	-
	Physical ID document	300.0 ^2^	-		300.0	-	300.0
	Document registration system	391.0 ^2^	-		391.0	-	391.0
Total	-	-	-		2291.8–4779.7	-	626.7–3073.3

^1^ Proportional cost of external management of plant residues; ^2^ excluding administrative fee. E: sanding; D: bare soil; EM: mechanical sanding. Source: own elaboration based on data provided by other authors [68,69], documents obtained from the Reinwaste project website [63,66,67] and consultations with specialized supply centers.

## Data Availability

The data presented in this study are available in this article.

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
