# Peer review of "The Management of Agriculture Plastic Waste in the Framework of Circular Economy. Case of the Almeria Greenhouse (Spain)"

_ijerph, 2021, doi:10.3390/ijerph182212042_

Round 1
Reviewer 1 Report
In this work, Castillo-Díaz et al. presented the current state and future perspectives on the generation and treatment of plastic waste in the intensive agriculture of Almeria, Spain.
The Title is good.
The Abstract, although straightforward, it could be refined into a more concise and appealing form.
Introduction
Agricultural irrigation depends on plastic, a known plastic additive that leaks from the material under natural UV and heat is bisphenol a, with known detrimental effects on human health, but as well as on plant and microbiota, this aspect should be included [1] along the introduction or at line 42-45 under plastic waste.
The overall quality of the work is good but the above mentioned need to be taken into consideration, another aspect is that due to the nature of the article, it is more suitable to be classified as an review.
References
1. Pop, C.-E.; Draga, S.; Măciucă, R.; Niță, R.; Crăciun, N.; Wolff, R. Bisphenol A Effects in
Aqueous Environment on Lemna minor. Processes 2021, 9, 1512.
https://doi.org/10.3390/pr9091512
Author Response
Please, see the attachment.

Reviewer 2 Report
The manuscript “The Management of Agriculture Plastic Waste in the Framework of Circular Economy. Case of the Almeria Greenhouse (Spain)” deals with a relevant environmental issue as it is greenhouse plastic management. From my point of view, a weakness of this work is that the data sources are referred but not accessible. Therefore, it would not be possible for other researchers to repeat this study from now in 5 years time. On the other hand, the manuscript is focussed just on the case of Almeria (Spain), which somehow limit the relevance of the study. In my opinion, the manuscript would benefit from the reference and comparison with other relevant Greenhouse locations around the world.
Next there are some comments that, from my point of view, need attention before this manuscript may be considered for publication.
Abstract
Please, correct separation by “,” and “.” in 1.716,6 kg·ha-1·year-1.
Section 3, page 5, line 185: Please, clarify “(an annual amount in the system of 37,147.4-57,347.4 t)”.
Section 3, page 5, lines 198-200: “The different types of polyethylene (LDPE, LLDPE and HDPE) and polypropylene made up the largest tonnage of production (96.9%) (Figure 1), being that these plastic materials pollute most of the ecosystems [30,77–79]”. This second part of this sentence is not understandable. Therefore, I suggest to divide this sentence into two, the first one from “The different…(Figure 1)”. Then, add a clear sentence, with the corresponding citations to give concise information.
Table 3: In the caption of the table, I guess that “by element” you mean “by type of polymer”. If so, please, correct. And also update the title of the column and replace “Compound” by “Polymer”. Also, values in the table should be revised, namely the use of “,” instead of “.” Finally, it should be clarified the basis of the values range (how did you obtain this range? Your source data showed such a dispersion? How many values did you have for each element? Would be possible to give results as mean ± SD?... ). Such a clarification may be done within the text or as a footnote.
Figure 1: Results presented in this figure are the same that were displayed in Table 3, aren’t they? If so, decide if you prefer to show them in a table or in a figure, but avoid repetition.
Section 3.2: Please, at the first citation of Table 4, use a sentence (avoid brackets) to refer what it shows.
Table 4, Table 5, Figure 2: Replace “,” by “.” for decimals separation.
Information within Section 3.4 needs to be related to the case study in Almeria (Spain). Please, improve the discussion so readers may follow the sequence and understand the pertinence of this section.
Conclusions should be summarized in order to give just main findings of the present research in a succinct way.
Author Response
Please, see the attachment.

Round 2
Reviewer 2 Report
Authors carried out a careful revision of their manuscript and submitted an improved version. From my point of view, the importance of this work is related to the topic, since published information is very scarce. However, the management of agriculture plastic is a very relevant environmental issue in some geographical regions, Almeria (Spain) being a good case study. In my opinion this manuscript is an interesting contribution for IJERPH and can be now accepted for publication.